# How can artificial intelligence boost firms' exports? evidence from China

Zhaozhong Zhang[1,2], Fangfang Deng[3]*

1 China Institute of Regulation Research, Zhejiang University of Finance and Economics, Hangzhou, Zhejiang, China, 2 The New Type Key Think Tank of Zhejiang Province "China Research Institute of Regulation and Public Policy", Hangzhou, Zhejiang, China, 3 College of Economics & Management, Zhejiang University of Water Resources and Electric Power, Hangzhou, Zhejiang, China

* sharondff@126.com

## Abstract

This paper explores the impact of artificial intelligence and industrial robots on firms' export behaviour and divides the impact mechanism into the productivity effect and labour substitution effect. It examines the effect of industrial robots on firms' export value by using Chinese Customs data, Chinese Industrial Firm data and robot data from the International Robot Federation (IRF). The main findings are as follows: Firstly, the impact of artificial intelligence and industrial robots on Chinese firms' export value is generally negative, which means the negative labour substitution effect dominates the positive productivity effect. Secondly, the impact of artificial intelligence varies significantly by industry, and the export value of firms from high-tech industries benefits from the use of industrial robots. Thirdly, the impact of artificial intelligence on firms' export value also varies by time; before 2003, the use of industrial robots showed mainly an inhibiting effect on firms' exports, which turned into a driving effect thereafter, and after 2006, industrial robots began to significantly promote firms' export. Finally, the higher the quality of export products, the more likely the use of industrial robots will be to promote firms' export value, and the higher the capital–labour ratio is, the more likely firms' export value will be to benefit from the use of artificial intelligence and industrial robots. On the basis of these findings, this study proposes promoting the productivity effect to dominate the labour substitution effect through technological progress and the improvement of export product quality.

## 1. Introduction

In recent years, China's export trade volume has achieved a year-on-year increase in conjunction with the growing level of opening up to the outside world. Furthermore, in the context of China's economy seeking high-quality development, the industrial structure is being continuously transformed and upgraded. Due to the gradual loss of the demographic dividend and the advantages of processing OEMs, China is gradually transforming from exporting resource- and labour-intensive products to exporting high-value-added products to achieve the transformation to comparative advantage.

Since 2000, China has gradually incorporated industrial robots into industrial production. In recent years, artificial intelligence (AI) and industrial robots have begun to be used widely

database can be accessed by contacting to National Bureau of Statistics of China and General Administration of Customs of China. The industrial robots data can be purchased from International Federation of Robotics (https://ifr.org/worldrobotics/). The author do not have special access privileges or permission to share the data. Although we cannot provide the raw data due to legal concerns, we provide the URL of our processed data which we have uploaded to baidu netdisk (https://pan.baidu.com/s/1vCwTe4xXUK2P2PuVAiu59w?pwd=8bh5). Considering we have provided the processed data underlying the findings. The interested researchers can replicate our study findings by directly obtaining the data and following the protocol in your Methods section. The authors have no special access privileges that others would not have on the data.

**Funding:** This study was supported by the National Social Science Fund of China (Grant No. 20FJLB009), Natural Science Foundation of Zhejiang Province (Grant No. LY20G030016) and the New Type Key Think Tank of Zhejiang Province "Research Institute of Regulation and Public Policy". The funders had no role in study design, data collection and analysis, decision to publish, or preparation of the manuscript.

**Competing interests:** The authors have declared that no competing interests exist.

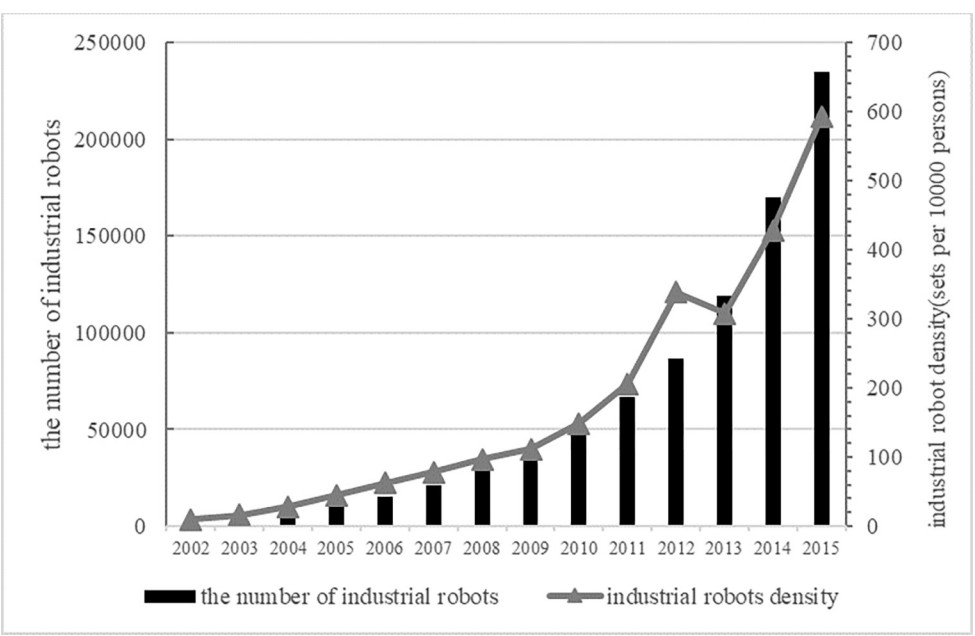

**Fig 1. The number and density of industrial robots in China from 2002 to 2015.** Source: World Robotics 2020. Industrial Robots of International Federation of Robotics.

in various sectors of China's manufacturing industry (as shown in Fig 1). On July 20, 2017, the State Council released the "New Generation of Artificial Intelligence Development Plan", proposing that the competitiveness of China's AI industry should reach a world-leading level by 2030. This proposal indicates that in the coming years, the development of AI in China will accelerate and will also more profoundly impact China's economic development, industrial structure upgrade, and export trade transformation. The use of artificial intelligence and industrial robots affects the productivity of micro-enterprises through labour substitution. According to the new-new trade theory, the use of AI will further affect the export decisions of enterprises (Melitz, 2003) [1]. Meanwhile, it has been argued in the literature that in the process of gradual diffusion of AI and industrial robots and the realisation of labour substitution, developing countries extract different benefits than developed countries do, whose productivity and capital accumulation endowments determine that they can better use the productivity gains brought by industrial robots and AI, which in turn widens the North–South gap (Alonso et al., 2020; Korinek and Stiglitz, 2021) [2,3]. Has China benefited from the global AI wave in the use of industrial robots and the substitution of low-end labour, or has the North–South gap been further widened by it? Can the use of AI and industrial robots promote Chinese firms' exports and transform their comparative advantage? As there is a relative lack of research on these questions, this paper explores the impact of the introduction of AI and industrial robots on the export behaviour of Chinese micro-enterprises from an open economy perspective.

 The contributions of this paper are as follows:

i. In this paper, the impact of AI on firms' export is divided into the productivity effect and labour substitution effect. This research perspective of identifying productivity and labour substitution effects is more conducive to simplifying the complex mechanism of artificial intelligence affecting firms' exports.

ii. Matching the industrial robot data at the industry level of the International Robotics Federation with Chinese industrial enterprise data and customs data, this paper constructs the micro-data basis of the impact of AI and industrial robots on firms' exports. The database used in this paper can be used as a data sample to study the impact of AI on the export behaviours of Chinese enterprises.

iii. Exploring the mechanism of the impact of the use of AI and industrial robot on firms' export behaviour, and then answering the question of which factors lead to different trade performances of heterogeneous firms after they start using industrial robots, provides a possible path for identifying how AI can better promote China's opening-up.

The remainder of this paper is organised as follows: Chapter 2 presents the literature review and theoretical hypotheses. Chapter 3 concerns materials and methods, including data and variables settings. Chapter 4 presents the empirical analysis and results, and Chapter 5 draws conclusions and makes policy recommendations.

## 2. Literature review and theoretical hypotheses

### 2.1 Literature review

The literature on the economic impact of AI has focused on AI's impact on productivity, economic growth, and the labour market. Moreover, some scholars have explored the impact of AI on income inequality based on these types of issues.

**2.1.1 The literature on the impact of AI on productivity and economic growth.** Brynjolfsson and McAfee (2014) referred to AI and the digital revolution as the second machine revolution, arguing that they are as important as the industrial revolution in driving development [4]. It has been widely recognised in the literature that the economic impact of AI is mainly reflected in the fact that the use of AI and robotics will increase productivity and thus promote economic growth.

Aghion et al. (2017) argued that there are two possible effects of AI and automation on economic growth: One is to increase productivity and thus increase share of return on capital by replacing labour, and the other is to lead to higher costs in the non-automated sector, reducing the share of return on capital [5].

The productivity boost caused by AI is closely related to its substitution of labour. Acemoglu and Restrepo (2018) attributed the driving force of technological progress to automation and the creation of new complex work tasks. Automation has both substitution and productivity effects; the former reduces labour demand and thus leads to lower wages, whereas the latter replaces labour with robots and thus reduces costs and increases productivity [6].

With the increased availability of data, a growing number of empirical studies examining the impact of AI or industrial robots on productivity have emerged. Kromann et al. (2011) found that automation contributes to productivity gains in both the short term and the long term [7]. Graetz and Michaels (2018) conducted an empirical analysis using industry panel data for 17 countries over the period 1993–2007 and found that the use of robots did contribute to economic growth by increasing productivity, with the use of industrial robots increasing the economic growth rate by approximately 0.37% [8]. Ballestar et al. (2020) used Spanish manufacturing data to verify the positive impact of robot use on firm productivity [9].

In addition, some scholars have argued that since the impact of AI on the share of capital returns is uncertain in the short term, when the share of capital returns reaches a steady state, the rate of economic growth brought about by AI will depend mainly on the rate of change in total factor productivity, so the long-term impact of AI on economic growth will depend on whether its impact on the rate of technological progress is short-term or long-term. In line

with this argument, some scholars have also investigated whether there is an economic singularity (i.e., after crossing that point in time, the economy will maintain sustained growth, and the rate of growth will continue to accelerate) in economic growth brought about by AI (Nordhaus, 2015; Aghion et al., 2017) [5,10].

**2.1.2 The literature on the impact of AI on the labour market.** Autor et al. (2003) first proposed the ALM model, which divides labour tasks into programmed tasks (routine tasks), which require only low-skilled labour, and non-programmed tasks, which require high-skilled labour. Moreover, it is believed that robots and automation can replace only low-skilled labour, not high-skilled labour, to complete programmed tasks [11]. Many scholars have extended and empirically tested the ALM model. Acemoglu and Restrepo (2017a) analysed the impact of industrial robots on the U.S. labour market between 1990 and 2007 and found that an increase of one robot per 1,000 people reduced the share of the employed population by about 0.18%–0.34% [12]. Acemoglu et al. (2020) conducted an empirical analysis using data from the French manufacturing sector and also concluded that robot use reduces employment and increases productivity [13].

Several studies have reached different conclusions on the question of whether the use of AI and robots necessarily reduces employment. Frey and Osborne (2013) extended the ALM model by assuming that non-programmed tasks also require both high-skilled and low-skilled labour inputs and found that the effect of AI and automation on the employment and wages of high-skilled workers under this assumption was uncertain [14]. Acemoglu and Restrepo (2018) suggested that AI and automation may reduce employment in traditional tasks, but they can also increase employment by creating new labour tasks [6]. Some scholars have also discovered that the use of robots does not significantly impact employment (Dauth et al., 2017; Graetz and Michaels, 2018) [8,15].

**2.1.3 The literature on the impact of AI on income inequality.** Some scholars have highlighted that AI, while boosting productivity for economic growth, may exacerbate income inequality, which is mainly triggered through the substitution of labour. Berg et al. (2016) argued that there are two main reasons for income inequality due to AI: First, the substitution of labour (especially low-skilled labour) by automated factors, such as robots, leads to the capital in total income share being elevated; that is, it causes differences in returns to different factors; second, wage inequality is increased due to the replacement of low-skilled labour by robots, resulting in further wage increases for skilled labour [16].

Benzell et al. (2015), as well as DeCanio (2016), have provided empirical evidence regarding the first impact path [17,18]. For the second impact path, Acemoglu and Restrepo (2017b) revised the assumptions of the ALM model by assuming that both low- and high-skilled labour may be replaced by robots to varying degrees and found that although the overall impact of AI and automation on wages is not clear, the substitution of low-skilled labour by robots does exacerbate wage inequality across different categories of labour, while the replacement of high-skilled labour by robots instead reduces wage inequality [19].

**2.1.4 The literature on the impact of AI in the open economy.** Regarding the economic impact of AI in an open economy, some scholars generally agree that the emergence of AI brings new opportunities for international trade while increasing productivity and promoting economic growth but that it may also exacerbate international inequality. By constructing a two-sector model, Alonso et al. (2020) found that due to the initial capital stock and productivity endowments differing, the introduction of AI and industrial robots may further worsen the terms of trade of developing countries, which in turn might widen the gap between developed and developing countries [2]. By classifying the use of AI and industrial robots as resource-saving and labour-substituting innovations, Korinek and Stiglitz (2021) found that when an economy has natural resource redundancy or labour redundancy, the introduction of AI may lead

to a deterioration in terms of the trade of that economy [3]. The above-mentioned related studies are all based on the assumption that AI and industrial robots will replace low-skilled labour, and the findings also point to the possibility that AI may exacerbate economic inequalities internationally. Scant literature has studied the trade impact of AI, with Goldfarb and Trefler (2018) noting that the scale economy of AI determines is more likely to be developed in countries with a large population and abundant data [20]. In addition, Goldfarb and Trefler (2018) discussed the effectiveness of strategic trade protection policies in the development of AI. It is important to note that AI, as discussed by Goldfarb and Trefler (2018), is emphasised more for its knowledge properties and that the production properties of robots are ignored.

In summary, it can be seen that the existing literature on the economic impact of AI and industrial robots focuses mostly on enterprise productivity and labour, and the substitution of labour by AI and industrial robots is the common starting point of the above studies, with the former starting from AI and industrial robots increasing enterprise productivity and thus promoting economic growth, and the latter exploring the impact of AI and robots on employment rates and wages based on the substitution of labour. Nevertheless, although some scholars have argued that AI has the potential to drive trade or change the pattern of international trade, there is little empirical evidence to verify this.

## 2.2 Theoretical hypotheses

On the basis of the existing literature, this paper assumes that the impact of AI and the use of industrial robots on firms' exports is reflected in two effects, namely the productivity effect and the substitution effect.

**2.2.1 Productivity effect.** Graetz and Michaels (2018) found that the use of industrial robots provides a substitution of low-skilled labour, realising labour productivity and total factor productivity gains [8]. Kromann et al. (2020) used industry panel data from nine countries and found that each unit increase in industrial robot density produces total factor productivity gains of 5% [7]. This result was further verified by Ballestar et al. (2020), who used Spanish data [9]. Melitz (2003) stated that micro-enterprise exporting is essentially a self-selective behaviour of high-productivity firms, with higher productivity firms tending to export products [1], while some empirical literature also verifies that the export volume of enterprises is positively correlated with their productivity levels (Zhang and Wang, 2020) [21]. Accordingly, this paper proposes that the productivity effect of AI affects the exports of micro-enterprises; that is, a positive transmission mechanism of increasing the total factor productivity of enterprises through the use of industrial robots and thus promoting their exports. According to the productivity effect, the use of AI is positively related to the export value of enterprises.

**2.2.2 Substitution effect.** A series of studies by Acemoglu and Restrepo (2017a, 2017b, 2018) demonstrate that the substitution of labour by industrial robots is mainly in the form of the substitution of low-skilled labour [6,12,19]. For high-technology industries, the majority of the workforce in the industry is highly skilled, so the workforce is less likely to be replaced by industrial robots, and the increase in robots correspondingly improves product quality and product competitiveness, thus promoting product exports. However, for non-high-technology industries, where low-skilled labour constitutes a significant proportion of workers, low-skilled labour is more likely to be replaced by industrial robots. Alonso et al. (2020) showed that the substitution of low-skilled labour by industrial robots leads to a decrease in the relative prices of products in the sector, which, while worsening the terms of trade, further leads to a decrease in investment in the sector and ultimately discourages firms from exporting [2]. Thus, for high-technology industries, the substitution effect of AI and industrial robots is reflected in a positive relationship between industrial robot density and firms' exports. In contrast, for non-

high-technology industries, the substitution effect is reflected in a negative relationship between the two.

On combining the two effects of AI and industrial robot use on firms' exports, it becomes clear that for high-tech industries, both the productivity effect and the substitution effect are positive, while for non-high-tech industries, the overall effect depends on which one is superior, the substitution effect or the productivity effect: If the substitution effect dominates the productivity effect, then AI and industrial robot use will generally discourage firms' exports, and if the productivity effect dominates the substitution effect, then AI and industrial robot use will generally promote exports. For a country or economy, if the productivity effect of AI and industrial robot use on industry-wide firms' exports is greater than the negative substitution effect, the country or economy will reflect the "developed country characteristic" of AI use. That is, AI and industrial robots will promote firms' exports. If the negative substitution effect is greater than the productivity effect, the country or economy will have a relatively low share of high-technology industries and exhibit "developing country characteristics"; that is, the use of AI and industrial robots will worsen the terms of trade by replacing low-skilled labour and thus discouraging exports.

Fig 2 illustrates the theoretical hypotheses. The productivity effect of AI and industrial robots promotes firms' exports by improving TFP. The substitution effect of industrial robots promotes firms' exports if they belong to high-tech industries. If the firms belong to non-high-tech industries, the substitution effect of industrial robots will reduce their exports. Therefore, for non-high-tech industries, the overall impact of AI and industrial robots on firms' exports depends on which effect, the productivity effect or the substitution effect, dominates.

The theoretical hypotheses are proposed:

H1: The overall impact of AI and industrial robots on Chinese firms' exports depends on which effect, the productivity effect or the substitution effect, dominates.

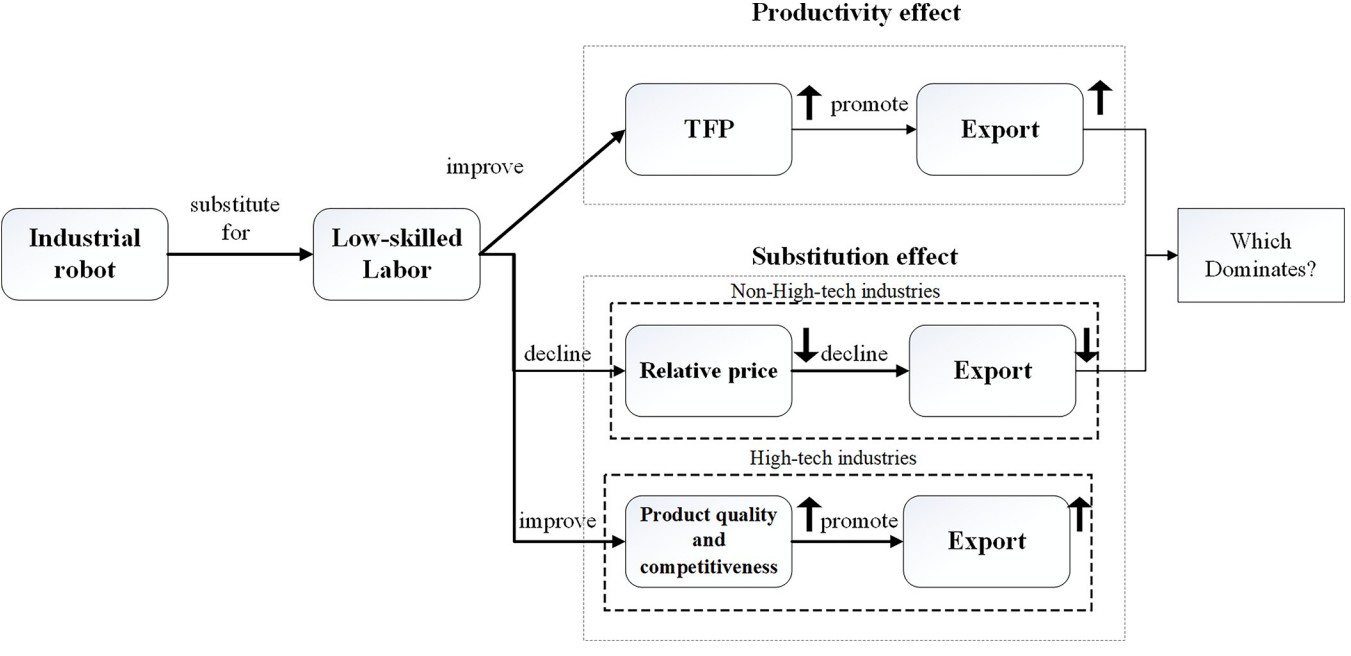

**Fig 2. The theoretical hypotheses.**

H2: The impact of AI and industrial robots varies significantly by industry, and the export
value of firms from high-tech industries benefits from the use of industrial robots.

In the following chapter, an empirical analysis is used to explore the characteristics of the
use of AI and industrial robots in China. Moreover, further analysis is performed to ascertain
which factors determine the productivity effect and the labour substitution effect.

## 3. Materials and methods

### 3.1 Data

The data used in this paper are from the following databases:

The first is the database of Chinese industrial enterprises, which contains data on state-
owned enterprises and non-state-owned manufacturing enterprises above the scale (with
annual sales over RMB 5 million).

The sample size of the database of Chinese industrial enterprises is large, and there are
some abnormal and invalid samples, so we first referred to Yu and Tian (2012) and Feenstra
et al. (2014) to eliminate the abnormal samples [22,23]: (i) Eliminate the samples without
major financial indicators, such as total assets, gross industrial output value, total fixed assets,
net fixed assets, paid-in capital, and main business income; (ii) exclude samples with missing
financial indicators; (iii) exclude samples with current assets, the total value of fixed assets, and
the net value of fixed assets greater than total assets in accordance with generally accepted
accounting principles (GAAP); (iv) exclude samples with accumulated depreciation less than
current-year depreciation; (v) exclude samples with fewer than eight employees; (vi) exclude
samples without an enterprise legal person code; (vii) eliminate samples with invalid establish-
ment time (established before January or after December).

Second, the Chinese Customs trade database provides monthly data on the imports and
exports of various types of HS8 code products by micro-enterprises to different destination
countries. Since the objective of this paper is to examine the export behaviour of enterprises,
only the export trade data was used, and, firstly, the monthly data was summed up into annual
data according to the HS four-digit code for subsequent processing. In addition, this paper
retained only the data on general trade exports for analysis.

Third, this study used industry-level industrial robot data from the International Robotic
Federation's (IRF) annual report on industrial robots. This data contains information on the
number of ISO-compliant industrial robots in 75 countries or regions and divides the use of
industrial robots worldwide by application, type, and industry in which they are used. The
data selected for this paper includes three dimensions–country, industry, and time–and spans
the period 2002–2013. The industrial robots data was matched to industrial enterprise data
and customs data by industry and year.

### 3.2 Variables settings

In this paper, the export value of micro-enterprises under the HS four-digit code was used as
the explanatory variable. The core explanatory variables are presented below.

**3.2.1 Industrial robot density.**   In this paper, the logarithm of the density of industrial
robots in use (also expressed as the number of industrial robots operating per unit of employ-
ment) was used to measure the level of AI in the industry.

**3.2.2 Total Factor Productivity (TFP).**   New-new trade theory suggests that firms' export
behaviour is a self-selection of high-productivity firms (Melitz, 2003) [1]; that is, high-produc-
tivity firms will prefer to export. Therefore, this paper added total factor productivity as a con-
trol variable to the empirical model. In this paper, both the OP and LP methods were used to

calculate TFP (Olley and Pakes, 1996; Levinsohn and Petrin, 2003) [24,25], where the OP method is used considering the missing variable in the database of Chinese industrial enterprises; the sample time span for calculating TFP with industrial value added as output Y is 2002–2007 (the indicator of industrial value added is missing after 2007); and the sample time span for calculating total factor productivity with gross industrial output value as output Y is 2002–2013. The estimated TFP time span is 2002–2007 using the LP method for estimating total factor productivity because the industrial value added and intermediate input indicators are needed. The capital input was calculated using the perpetual inventory method, and the number of employees was taken as the labour input. When the OP method was used to estimate TFP, the capital stock was estimated at a 5% depreciation rate (Wang and Fan, 2000) [26], a 10% depreciation rate (Zhang et al., 2004; Shan, 2008) [27,28], and a 15% depreciation rate (Zhang et al., 2009) [29].

**3.2.3 Importing country characteristics control variables.** On the basis of the classical trade gravity model, two variables–tariffs of importing countries and the size of the economy of importing countries (constant price GDP)–were added in this paper to reflect the influence of demand-side factors on firms' export behaviour in China. For tariff data, the weighted average tariff data of HS four-digit codes from the World Bank TRAINS trade system was used, and the importing country GDP data was obtained from the CEPII-BACI database. The tariff data was then matched to the original data by year, product category, and importing country, and the GDP data of the importing country was matched to the original data by country and year. It was assumed that importing country tariffs are negatively correlated with firms' exports and that the importing country's economic size is positively correlated with firms' exports.

**3.2.4 Other firm characteristic variables.** According to Fontagné et al. (2015), firm size affects its export value, and larger firms are more inclined to export [30], so this paper used total firm assets as an indicator of firm size and used it as a control variable for firm characteristics.

Other firm and importing country characteristics were controlled for by adding firm-fixed effects, product-fixed effects, and importing-country-fixed effects. Table 1 reports the descriptive statistics of the variables.

**Table 1. Descriptive statistics of the variables.**

| Variables | mean | SD | maximum | minimum | samples |
|---|---|---|---|---|---|
| Value of export | 438403 | 14954772 | 16943636480 | 0 | 9174929 |
| TFP (OP Method, industrial value as Y, 5%) | 3.645 | 0.913 | 10.54 | -6.702 | 2860794 |
| TFP (OP Method, industrial value as Y, 10%) | 3.780 | 0.922 | 11.72 | -6.501 | 2834263 |
| TFP (OP Method, industrial value as Y, 15%) | 3.889 | 0.933 | 10.61 | -6.317 | 2788938 |
| TFP (OP Method, gross industrial output as Y, 5%) | 5.766 | 0.996 | 12.62 | -7.133 | 8332796 |
| TFP (OP Method, gross industrial output as Y, 10%) | 6.482 | 1.013 | 13.48 | -6.144 | 7858191 |
| TFP (OP Method, gross industrial outpu as Y, 15%) | 6.075 | 0.997 | 13.01 | -6.832 | 7549480 |
| TFP (LP Method) | 7.272 | 1.179 | 12.77 | -2.638 | 2860794 |
| Tariff | 6.019 | 9.905 | 1815 | 0 | 8530032 |
| assets | 308680 | 2869772.710 | 100447392 | 0 | 9174929 |
| GDP of importing countries | 2394568592 | 4005557588.007 | 16270135296 | 78900 | 9019530 |
| Capital-labor ratio | 64.81 | 613.9 | 424535 | -328.0 | 8987878 |
| Quality of export products | 0.517 | 0.179 | 1 | 0 | 8980252 |

Note: 5%, 10%, and 15% in Table 1 mean the capital input was calculated using the perpetual inventory method, and the capital stock was estimated at a 5%, 10%, or 15% depreciation rate, respectively.

## 4. Empirical analysis and results

### 4.1 Benchmark regression

This paper first conducted OLS benchmark regressions on the effects of AI and firms' total factor productivity on their exports. The explanatory variables are the logarithm of firms' exports of HS4 code products to a destination in the current year, and the logarithm of robot density and total factor productivity of firms are the core explanatory variables. Moreover, control variables such as tariffs of importing countries, GDP of importing countries, and firm size measured by total assets were added. In the benchmark regression, the OP method was used to measure TFP with industrial value added as the output variable, and the capital stock was estimated using the perpetual inventory method using a 10% depreciation rate. The regression results are shown in the first two columns of Table 2.

In Table 2, Column 1 shows the regression results with the inclusion of only three variables–robot density, total factor productivity, and tariff–while Column 2 reports the regression results, including the control variables of firm size and GDP of the importing countries. According to the regression results reported in Table 2, the robot density is significantly negatively correlated with the firms' export value, and the total factor productivity is significantly positively correlated with the export value. The larger the firm and the larger the GDP of the importing countries are, the more the firm exports. However, the OLS benchmark regression results indicate that tariffs are significantly positively correlated with the export value, which we believe is the result of omitted variables. To address the endogeneity and estimation bias caused by the omitted variables, we added year-fixed effects to control for all variables that simply vary over time, firm-fixed effects to control for other firm characteristics variables, and

**Table 2. The results of the benchmark regression.**

| Variable | (1) OLS1 | (2) OLS2 | (3) FE1 | (4) FE2 |
|---|---|---|---|---|
| Robot density | -0.060*** | -0.066*** | -0.094*** | -0.087*** |
| | (-81.381) | (-89.297) | (-2.938) | (-2.718) |
| TFP | 0.162*** | 0.023*** | 0.128*** | 0.126*** |
| | (25.701) | (3.537) | (12.209) | (12.052) |
| Tariff | 1.294*** | 1.718*** | -0.138*** | -0.123*** |
| | (55.677) | (73.883) | (-4.802) | (-4.251) |
| Firm Size | | 0.137*** | | 0.142*** |
| | | (126.912) | | (21.323) |
| GDP | | 0.162*** | | 0.188*** |
| | | (147.798) | | (7.327) |
| Year FE | No | No | Yes | Yes |
| Importing country FE | No | No | Yes | Yes |
| Product FE | No | No | Yes | Yes |
| Industry FE | No | No | Yes | Yes |
| Firm FE | No | No | Yes | Yes |
| Samples | 2172256 | 2165325 | 2163394 | 2156456 |
| $R^2$ | 0.006 | 0.022 | 0.335 | 0.335 |
| Adjusted $R^2$ | 0.006 | 0.022 | 0.316 | 0.316 |
| $F$-value | 4017.733 | 9684.458 | 59.478 | 138.015 |

Note:

*, **, and *** indicate significance at the significance levels of 10%, 5%, and 1%, respectively; there is no further discussion on this in the subsequent text.

importing-country-fixed effects to control for demand-side characteristics variables of importing countries other than importing country tariffs and GDP, in addition to product-fixed effects and industry-fixed effects. The model is shown below.

$$Lnexport_{ejkt} = \beta_0 + \beta_1 Lnrobot_{it} + \beta_2 Lntfp_{et} + \beta_3 LnTariff_{jkt} + \beta_4 LnGDP_{jt} + \beta_5 Lnassets_{et} + v_t$$
$$+ \mu_j + \eta_k + \varphi_e + \xi_i + \varepsilon_{ejkt}, \tag{1}$$

where $Lnexport_{ejkt}$ denotes the logarithm of the export value of product $k$ of firm $e$ to destination importing country $j$ in year $t$. $Lnrobot_{it}$ is the logarithm of robot density of industry $i$ in China in year $t$. $Lntfp_{et}$ refers to the total factor productivity of firm $e$ in year $t$. $Lnassets_{et}$ denotes the total assets of firm $e$ in year $t$, and $LnTariff_{jkt}$ t indicates the logarithm of the tariff rate imposed by importing country $j$ on product $k$ in year $t$. To overcome the estimation bias caused by zero tariffs, this study adopted the method of adding 100 to the tariffs of all importing countries and then taking the logarithm. $LnGDP_{jt}$ denotes the logarithm of the GDP of importing country $j$ in year $t$. In addition, $v_t$ is the time-fixed effect; $\mu_j$ is the importing-country-fixed effect; $\eta_k$ is the product-fixed effect; $\varphi_e$ is the firm-fixed effect; $\xi_i$ is the industry-fixed effect, and $\varepsilon_{ejkt}$ is the random error term.

The regression results after the inclusion of fixed effects are shown in Columns 3 and 4 in Table 2, where the tariff coefficient is significantly negative after the inclusion of importing-country- and product-fixed effects, suggesting that tariffs in importing countries discourage enterprises from exporting by increasing trade costs, and furthermore, according to the results reported in Table 2, firms with higher productivity and of larger size are correspondingly higher in export value. In addition, the density of industrial robots is still significantly negatively correlated with the export value after the inclusion of fixed effects, indicating that, overall, the use of industrial robots in China was not able to promote firms' exports in the period 2002–2013. Is the use of AI and industrial robots simply a disincentive for firms' exports? Do different industries and heterogeneous firms behave differently in the use of industrial robots? This paper further investigates these questions.

## 4.2 Robustness tests

For a robustness check, we estimated the benchmark regression model using different ways of estimating total factor productivity separately. In Table 3, the first two columns adopt the OP method and estimate total factor productivity using industrial value added as the output variable. Since the industrial-value-added data provided in the industrial enterprise data is only from before 2008, the first two columns of the regression also span the period 2002–2007, and the difference between the first two columns is that the capital stock is estimated using the perpetual inventory method using 5% and 15% depreciation rates, respectively. Columns 3–5 use the OP method and estimate total factor productivity using gross industrial output (GIP) as the output variable, with data spanning 2002–2013, and Columns 3, 4, and 5 use 5%, 10%, and 15% depreciation rates, respectively, to calculate the capital stock. The last column uses the LP method and estimates total factor productivity, still spanning the period 2002–2007.

According to the results reported in Table 3, the total factor productivity of enterprises measured using different approaches is still significantly and positively correlated with exports; tariff rates are significantly and negatively correlated with exports; the larger the size of the enterprise, the more it exports, and GDP of the importing country is also significantly and positively correlated with firms' exports. It was also found that industrial robot density is significantly negatively correlated with firms' exports when total factor productivity is estimated using industrial value added as an output variable, while the coefficient of the industrial robot density variable is positive when total factor productivity is estimated using gross industrial

**Table 3. Robustness test results.**

| Variable | (1) op5 | (2) op15 | (3) op5_1 | (4) op10_1 | (5) op15_1 | (6) lp |
|---|---|---|---|---|---|---|
| Robot density | -0.090*** | -0.080** | 0.005 | 0.008 | 0.012* | -0.090*** |
| | (-2.824) | (-2.473) | (0.725) | (1.180) | (1.743) | (-2.809) |
| TFP | 0.128*** | 0.115*** | 0.419*** | 0.534*** | 0.377*** | 0.330*** |
| | (12.818) | (10.837) | (38.204) | (39.207) | (30.537) | (14.294) |
| Tariff | -0.121*** | -0.125*** | -0.103*** | -0.114*** | -0.114*** | -0.121*** |
| | (-4.199) | (-4.277) | (-5.631) | (-6.074) | (-5.934) | (-4.202) |
| Firm Size | 0.139*** | 0.138*** | 0.083*** | 0.088*** | 0.096*** | 0.125*** |
| | (21.379) | (20.340) | (28.671) | (28.482) | (30.109) | (18.904) |
| GDP | 0.183*** | 0.184*** | 0.019** | 0.025*** | 0.029*** | 0.180*** |
| | (7.153) | (7.126) | (2.074) | (2.760) | (3.163) | (7.047) |
| Year FE | Yes | Yes | Yes | Yes | Yes | Yes |
| Importing country FE | Yes | Yes | Yes | Yes | Yes | Yes |
| Product FE | Yes | Yes | Yes | Yes | Yes | Yes |
| Industry FE | Yes | Yes | Yes | Yes | Yes | Yes |
| Firm FE | Yes | Yes | Yes | Yes | Yes | Yes |
| Samples | 2176560 | 2121057 | 6693180 | 6309448 | 6056363 | 2180704 |
| $R^2$ | 0.335 | 0.336 | 0.334 | 0.334 | 0.335 | 0.335 |
| Adjusted $R^2$ | 0.316 | 0.316 | 0.321 | 0.322 | 0.322 | 0.316 |
| $F$-value | 143.232 | 122.952 | 475.459 | 506.347 | 379.965 | 151.540 |

output value as an output variable. Since there is a significant difference in the sample time span between the two estimation methods, there may be significant temporal differences in the effects on the export behaviour of micro-enterprises, which are also verified in the following sections.

## 4.3 The regression results of different industries

The theoretical hypotheses of this paper assume that the impact of AI and industrial robots on firms' exports is the result of the combined effect of productivity and substitution effects, and there is a significant difference between the positive productivity effect and the negative substitution effect for different types of industries. On the basis of these theoretical hypotheses, we conducted regression analysis for different types of industries separately.

According to the classification of "high-tech industry" of the Chinese National Bureau of Statistics, the high-tech industry is defined in this paper as a manufacturing industry with relatively high R&D investment intensity, including pharmaceutical manufacturing, aviation, spacecraft and equipment manufacturing, electronic and communication equipment manufacturing, computer and office equipment manufacturing, medical equipment and instrumentation manufacturing, information chemicals manufacturing, and six other categories. After dividing the industries into high-tech and non-high-tech industries, we conducted empirical tests on the mechanisms of industrial robot use and total factor productivity affecting firms' exports in these two major categories of industries; the regression results are shown in Table 4.

The first three columns in Table 4 report the regression results for high-technology industries. Column 1 reports the regression results for 2002–2007 data using the OP method to estimate total factor productivity with industrial value added as the output variable. Column 2 reports the regression results for 2002–2013 data using the OP method to estimate total factor

**Table 4. The regression results of different industries.**

| Variable | High-tech industries | | | Non-high-tech industries | | |
|---|---|---|---|---|---|---|
| | (1) | (2) | (3) | (4) | (5) | (6) |
| | op10 | op10_1 | lp | op10 | op10_1 | lp |
| Robot density | 0.113** | 0.152*** | 0.116** | -0.219*** | 0.016* | -0.223*** |
| | (2.383) | (7.728) | (2.466) | (-4.223) | (1.764) | (-4.338) |
| TFP | 0.146*** | 0.553*** | 0.450*** | 0.120*** | 0.519*** | 0.300*** |
| | (7.484) | (20.325) | (9.930) | (9.570) | (32.760) | (11.131) |
| Tariff | -0.025 | -0.028 | -0.024 | -0.284*** | -0.246*** | -0.282*** |
| | (-0.365) | (-0.671) | (-0.351) | (-8.647) | (-11.529) | (-8.624) |
| Firm Size | 0.178*** | 0.098*** | 0.151*** | 0.131*** | 0.090*** | 0.118*** |
| | (13.062) | (16.030) | (11.095) | (17.300) | (25.056) | (15.628) |
| GDP | 0.325*** | 0.139*** | 0.315*** | 0.120*** | -0.024** | 0.112*** |
| | (6.532) | (8.199) | (6.357) | (4.035) | (-2.268) | (3.792) |
| Year FE | Yes | Yes | Yes | Yes | Yes | Yes |
| Importing country FE | Yes | Yes | Yes | Yes | Yes | Yes |
| Product FE | Yes | Yes | Yes | Yes | Yes | Yes |
| Industry FE | Yes | Yes | Yes | Yes | Yes | Yes |
| Firm FE | Yes | Yes | Yes | Yes | Yes | Yes |
| Samples | 624021 | 2044835 | 630810 | 1532304 | 4263892 | 1549763 |
| $R^2$ | 0.357 | 0.364 | 0.357 | 0.338 | 0.331 | 0.338 |
| Adjusted $R^2$ | 0.338 | 0.351 | 0.337 | 0.319 | 0.318 | 0.319 |
| $F$-value | 56.448 | 168.391 | 63.261 | 102.094 | 386.039 | 111.517 |

productivity with gross industrial output as the output variable. Column 3 presents the regression results for the 2002–2007 data using the LP method to estimate total factor productivity, and the last three columns present the regression results for non-high-tech industries. As can be seen from Table 4, the main difference between high-tech and non-high-tech industries is reflected in the effect of AI and industrial robot use on firms' exports, with a significant positive correlation between firms' exports and industrial robot density for high-tech industries, and a significant negative correlation between firms' exports and industrial robot density for non-high-tech firms overall, except for Column 5 for the time span 2002–2013. The coefficient of robot density is positive with data spanning 2002–2013, which again reflects that the impact of AI on the export behaviour of micro-enterprises in China has some temporal differences. We empirically verified the theoretical hypotheses in Table 4 and found that for high-tech industries, the use of AI and industrial robots generally promotes firms' exports because the productivity and substitution effects are both positive. In contrast, for China's non-high-tech industries, the negative substitution effect is larger than the productivity effect, and the use of AI and industrial robots generally inhibits firms' exports.

## 4.4 The effect of AI and industrial robots on firms' exports with different time intervals

It was found in the previous empirical analysis that the impact of AI and industrial robots on firms' exports in China differed not only between industries but also over time, and when the time span of the sample was extended from 2002–2007 to 2002–2013, the density of industrial robots in the whole industry and non-high-tech industries began to show a positive correlation with enterprise exports. Accordingly, we first conducted a threshold panel regression using year as the threshold variable using Wang's (2015) method [31]. The threshold panel model is

**Table 5. The results of the threshold value test.**

| Model | Threshold | Upper Threshold | Lower Threshold | RSS | MSE | F-value | P-value |
|---|---|---|---|---|---|---|---|
| Single Threshold Model | 2003 | 2002 | 2004 | 6473 | 1.194 | 170.8 | 0 |

as follows:

$$lnexport_{ejkt} = \alpha + \beta_1 lnrobot_{it}I(Diver \leq \gamma_1) + \beta_2 lnrobot_{it}I(Diver > \gamma_1) + X_{et}\gamma' + \mu lnY_{jkt} + \varepsilon_{ejkt}, \quad (2)$$

where $Diver$ is the threshold variable, which in this paper is the annual variable; $\gamma_1$ represents a specific threshold value; and $I$ is an indicator function with a value of 1 when the condition is met and 0 otherwise. The regression results are shown in Tables 5 and 6.

As can be seen from Tables 5 and 6, the $p$ values show significant rejection of the original hypothesis, and the threshold panel regression model supports the existence of a single threshold, with an annual threshold of 2003. However, considering that the original data needs to be processed into balanced panel data before conducting the threshold panel regression, thus losing a large number of samples, which may lead to estimation bias, we conducted the regression again in time intervals according to the threshold setting, and the results are shown in Table 7.

The first two columns of Table 7 report the results of regressions using 2003 as the threshold. They were used mainly to verify whether there is indeed a threshold effect on the impact of industrial robots on firms' exports. The regression results show that robot density is negatively related to firms' exports in the all-industry sample regression before 2003, while from 2003, the use of AI and industrial robots started to have a positive impact on firms' exports, which again verifies that there are time differences in the impact of AI on firms' exports. It was verified previously that firms' exports in China's high-tech industries are significantly and positively influenced by the use of AI and industrial robots, and to further verify the starting year of this positive promotion effect, we conducted further regressions on high-tech industries by time intervals, as shown in Table 7. Before 2004, the use of AI and industrial robots in China's high-tech industries showed a negative effect. After 2004, the use of industrial robots began to have a positive, although not significant, effect on firms' exports. Since 2007, the use of industrial robots in China's high-tech industry has had a significant positive effect on firms' exports.

**Table 6. The results of threshold regression.**

| Variable | Coefficient | Standard error | T-value | P-value | 95% confidence interval | |
|---|---|---|---|---|---|---|
| Tariff | -0.877 | 0.555 | -1.580 | 0.114 | -1.965 | 0.211 |
| TFP | 1.523 | 0.150 | 10.17 | 0 | 1.229 | 1.816 |
| Firm size | 0.552 | 0.0446 | 12.39 | 0 | 0.465 | 0.640 |
| GDP | 0.617 | 0.157 | 3.930 | 0 | 0.309 | 0.924 |
| Year threshold | | | | | | |
| Robot density before threshold | 0.177 | 0.0186 | 9.520 | 0 | 0.141 | 0.214 |
| Robot density after threshold | 0.0193 | 0.0188 | 1.030 | 0.304 | -0.0176 | 0.0562 |
| constant | -4.964 | 4.631 | -1.070 | 0.284 | -14.04 | 4.116 |
| *sigma u* | 1.686 | | | | | |
| *sigma e* | 1.140 | | | | | |
| *rho* | 0.686 | | | | | |
| *F*-value | 25.06 | | | | | |

Table 7. The results of regression with different time intervals.

| Variable | Single threshold (all-industry sample) | | | | regression with different time intervals (high-tech industries) | | | |
|---|---|---|---|---|---|---|---|---|
| | (1) | (2) | (3) | (4) | (5) | (6) | (7) | (8) |
| | 2002–2003 | 2004–2013 | 2002–2003 | 2002–2004 | 2002–2005 | 2002–2006 | 2002–2007 | 2002–2008 |
| Robot density | -0.074 | 0.012* | -0.147 | -0.027 | 0.038 | 0.057 | 0.165*** | 0.257*** |
| | (-0.980) | (1.686) | (-0.777) | (-0.304) | (0.638) | (1.209) | (4.048) | (7.028) |
| TFP | 0.592*** | 0.529*** | 0.320* | 0.542*** | 0.588*** | 0.695*** | 0.742*** | 0.683*** |
| | (6.202) | (36.971) | (1.675) | (4.305) | (6.983) | (10.533) | (13.471) | (14.749) |
| Tariff | 0.118** | -0.136*** | -0.052 | -0.074 | -0.048 | -0.081 | -0.035 | -0.047 |
| | (2.133) | (-6.738) | (-0.431) | (-0.766) | (-0.589) | (-1.129) | (-0.552) | (-0.863) |
| Firm Size | 0.072*** | 0.080*** | 0.013 | 0.114*** | 0.163*** | 0.175*** | 0.174*** | 0.154*** |
| | (2.969) | (24.603) | (0.244) | (3.509) | (7.947) | (11.700) | (14.309) | (15.294) |
| GDP | -0.045 | 0.022** | -0.299 | -0.236 | 0.082 | 0.106** | 0.302*** | 0.309*** |
| | (-0.283) | (2.107) | (-1.017) | (-1.596) | (1.127) | (1.965) | (6.973) | (9.183) |
| Year FE | Yes | Yes | Yes | Yes | Yes | Yes | Yes | Yes |
| Importing country FE | Yes | Yes | Yes | Yes | Yes | Yes | Yes | Yes |
| Product FE | Yes | Yes | Yes | Yes | Yes | Yes | Yes | Yes |
| Industry FE | Yes | Yes | Yes | Yes | Yes | Yes | Yes | Yes |
| Firm FE | Yes | Yes | Yes | Yes | Yes | Yes | Yes | Yes |
| Samples | 356381 | 5951343 | 107232 | 235119 | 354475 | 522819 | 737124 | 998352 |
| $R^2$ | 0.361 | 0.337 | 0.397 | 0.384 | 0.381 | 0.369 | 0.368 | 0.369 |
| Adjusted $R^2$ | 0.324 | 0.324 | 0.359 | 0.353 | 0.357 | 0.348 | 0.349 | 0.352 |
| F-value | 11.072 | 424.718 | 0.982 | 7.273 | 24.137 | 54.954 | 96.794 | 124.790 |

## 4.5 Endogeneity problems

This paper mainly overcomes the endogeneity problem caused by omitted variables by adding firm-fixed, year-fixed, importing-countries-fixed, and industry-fixed effects. While considering the possible endogeneity problem caused by causal inversion (i.e., firms that export more tend to choose to use industrial robots), this paper mainly constructed an instrumental variable for industrial robot density in China based on the construction method of the Bartik instrumental variable using U.S. industrial robot density of the corresponding industry.

The Bartik instrumental variable is considered to have good instrumental variable properties because the estimates are highly correlated with the actual values but not with the other residual terms (Bartik, 1991) [32]. In this paper, the instrumental variables were constructed as follows:

$$Instrument_{uit} = \frac{GP_{uit}}{GP_{cit}} \times \frac{Robot_{uit}}{EMP_{uit}}. \tag{3}$$

In Eq 3, $Robot_{uit}$ denotes the number of industrial robots in industry $i$ in the United States in year $t$. $EMP_{uit}$ denotes the number of people employed in industry $i$ in the United States in year $t$, and $\frac{Robot_{uit}}{EMP_{uit}}$ is the density of industrial robots in industry $i$ in the United States in year $t$. $GP_{uit}$ and $GP_{cit}$ are, respectively, the gross product of industry $i$ in the United States and China in year $t$. This paper further corrects the industrial robot density of U.S. industry $i$ in year $t$ by using the ratio of the share of U.S. industry $i$'s GDP to that of the Chinese industry's GDP in year $t$ and uses it as an instrumental variable for China's industrial robot density. The reasons for using the corrected U.S. industrial robot density as an instrumental variable for China's industrial robot density are as follows: First, although the U.S. industrial robot stock is ahead

**Table 8. The results of IV regressions.**

| Variable | All-industry sample | | | High-tech industries | | |
|---|---|---|---|---|---|---|
| | (1) | (2) | (3) | (4) | (5) | (6) |
| | op10 | op10_1 | lp | op10 | op10_1 | lp |
| Robot density | -0.018 | 0.085*** | -0.015 | 0.082*** | 0.114*** | 0.078*** |
| | (-1.220) | (10.659) | (-0.991) | (5.097) | (9.761) | (4.971) |
| TFP | 0.135*** | 0.604*** | 0.365*** | 0.192*** | 0.558*** | 0.506*** |
| | (7.457) | (32.778) | (9.238) | (3.302) | (15.929) | (3.966) |
| Tariff | 0.522*** | 0.321*** | 0.518*** | -0.008 | -0.307*** | -0.018 |
| | (15.600) | (17.265) | (15.585) | (-0.086) | (-7.387) | (-0.200) |
| Firm Size | 0.126*** | 0.057*** | 0.110*** | 0.044 | 0.028*** | 0.033 |
| | (9.765) | (14.192) | (8.597) | (1.064) | (3.556) | (0.792) |
| GDP | 0.199*** | 0.215*** | 0.199*** | 0.182*** | 0.191*** | 0.182*** |
| | (129.072) | (269.918) | (129.719) | (53.214) | (125.034) | (53.365) |
| Year FE | Yes | Yes | Yes | Yes | Yes | Yes |
| Importing country FE | Yes | Yes | Yes | Yes | Yes | Yes |
| Product FE | Yes | Yes | Yes | Yes | Yes | Yes |
| Industry FE | Yes | Yes | Yes | Yes | Yes | Yes |
| Firm FE | Yes | Yes | Yes | Yes | Yes | Yes |
| Samples | 1181810 | 4909304 | 1196985 | 273647 | 1602278 | 277188 |

of China's, the development trend in the sample coverage time is closer to China's, and the corrected U.S. industrial robot density according to the gross product has a high correlation with China's industrial robot density. Second, the U.S. industrial robot quantity and density are strictly exogenous to China's exports and other economic factor variables, and using them as instrumental variables helps mitigate the endogeneity problem of the model. The results of the regressions using instrumental variables are shown in Table 8.

The first three columns of Table 8 report the results of instrumental variable estimation for the all-industry sample, where Columns 1, 2, and 3 use total factor productivity estimated by the OP method with industrial value added as the output variable, total factor productivity estimated by the OP method with gross industrial output as the output variable, and total factor productivity estimated by the LP method as the output variable, respectively. The latter three columns report the results of estimating instrumental variables for high-technology industries. As seen in Table 8, after the instrumental variable of industrial robot density in the United States is used, the use of industrial robots in the all-industry sample is negatively correlated with firms' exports, except for Column 2, which is largely consistent with the previous regression results. The use of industrial robots is positively correlated with firms' exports in high-tech industries, which also reaffirms the robustness of the previous benchmark regression results.

In addition, to address the possible endogeneity problem of causal inversion, we also used robot density one year lagged as the IV and estimated the effect of AI and industrial robots on firms' exports using the GMM method. The regression results are shown in Table 9.

Consistent with Table 8, the first three columns of Table 9 still report the estimation results for the all-industry sample, while the last three columns report the estimation results for the high-technology industry. As can be seen from Table 9, the use of industrial robots is positively correlated with firms' exports in both the all-industry sample and high-tech industries after lagging the industrial robot density by one period, which is basically consistent with the previous benchmark regression.

**Table 9. Regression results with robot density one year lagged.**

| Variable | All-industry sample | | | High-tech industries | | |
|---|---|---|---|---|---|---|
| | (1) | (2) | (3) | (4) | (5) | (6) |
| | op10 | op10_1 | lp | op10 | op10_1 | lp |
| Robot density | 0.051*** | 0.078*** | 0.051*** | 0.048*** | 0.006* | 0.050*** |
| | (20.096) | (49.678) | (20.395) | (8.291) | (1.662) | (8.673) |
| TFP | 0.121*** | 0.460*** | 0.370*** | -0.075*** | 0.011 | -0.098* |
| | (8.701) | (30.340) | (12.818) | (-2.962) | (0.393) | (-1.744) |
| Tariff | 1.102*** | 1.043*** | 1.089*** | 0.405*** | 0.603*** | 0.384*** |
| | (23.544) | (39.024) | (23.456) | (3.696) | (9.804) | (3.525) |
| Firm Size | 0.193*** | 0.129*** | 0.170*** | 0.198*** | 0.131*** | 0.202*** |
| | (77.497) | (95.796) | (54.731) | (43.829) | (54.553) | (33.432) |
| GDP | 0.241*** | 0.243*** | 0.241*** | 0.212*** | 0.212*** | 0.211*** |
| | (104.084) | (194.027) | (104.690) | (47.247) | (89.162) | (47.364) |
| Year FE | Yes | Yes | Yes | Yes | Yes | Yes |
| Importing country FE | Yes | Yes | Yes | Yes | Yes | Yes |
| Product FE | Yes | Yes | Yes | Yes | Yes | Yes |
| Industry FE | Yes | Yes | Yes | Yes | Yes | Yes |
| Firm FE | Yes | Yes | Yes | Yes | Yes | Yes |
| Samples | 474385 | 1881418 | 481222 | 142373 | 605822 | 144065 |
| $R^2$ | 0.047 | 0.041 | 0.047 | 0.068 | 0.065 | 0.067 |
| Adjusted $R^2$ | 0.047 | 0.041 | 0.047 | 0.068 | 0.065 | 0.067 |
| F-value | 2320.381 | 8024.429 | 2344.391 | 886.427 | 3394.725 | 879.564 |

## 4.6 Analysis of the mechanism of AI and industrial robots affecting firms' exports

In the previous sections, we conducted a general validation of AI and industrial robot use affecting the export behaviour of Chinese micro-enterprises and found that the use of industrial robots somewhat inhibits firms' exports under the all-industry sample, but for high-tech industries, the use of industrial robots can play a role in promoting firms' exports. In the theoretical hypotheses, we attribute this finding to the question of which dominates, the productivity effect or the labour substitution effect. In the following sections, we further analyse the mechanism by which AI and industrial robots affect firms' exports, firstly, to verify the existence of a positive productivity effect and then to answer the question of which factors determine which of the two effects dominates.

**4.6.1 Validation of productivity effect.** The theoretical hypotheses of this paper suggest that the productivity effect is mainly reflected in the increase of total factor productivity of enterprises through the use of industrial robots, which affects the export behaviour of enterprises through the self-selection mechanism of exports of high-productivity enterprises, and in the previous empirical study part, we also verified that there is a significant positive correlation between total factor productivity of enterprises and their export value. Therefore, in the following paragraphs, we consider the role of productivity in the transmission mechanism of industrial robot density-firm total factor productivity-firm export value as a mediating effect and test it further. First, the mechanism by which industrial robot use affects total factor productivity is empirically tested. Then, to avoid estimation bias caused by the omitted variable problem, year, importing countries, product, firm, and industry fixed effects are included. The regression results are shown in Table 10.

**Table 10.  Regression results of the effect of industrial robot use on TFP.**

|  | (1) | (2) | (3) |
|---|---|---|---|
| **Variable** | *op10* | *op10_1* | *lp* |
| Robot density | 0.104*** | 0.003*** | 0.034*** |
|  | (45.986) | (14.988) | (30.894) |
| Year FE | Yes | Yes | Yes |
| Importing country FE | Yes | Yes | Yes |
| Product FE | Yes | Yes | Yes |
| Industry FE | Yes | Yes | Yes |
| Firm FE | Yes | Yes | Yes |
| Samples | 2304191 | 6804300 | 2329774 |
| $R^2$ | 0.742 | 0.829 | 0.856 |
| Adjusted $R^2$ | 0.735 | 0.826 | 0.852 |
| *F*-value | 2114.667 | 224.628 | 954.442 |

Column 1 of Table 10 reports the estimation results for the time span 2002–2007, with industrial value added as the output variable and total factor productivity estimated by the OP method. Column 2 reports the estimation results for the time span 2002–2013, with gross industrial output as the output variable and productivity estimated by the OP method. Column 3 corresponds to total factor productivity estimated by the LP method, still spanning the period 2002–2007. In Table 10, the regression results show that firms' productivity does increase significantly by introducing industrial robots after several fixed effects have been controlled, thus validating the first link of the productivity effect.

We further examined the mediating effect of total factor productivity on the use of industrial robots affecting firms' exports. The model estimated both the effect of the explanatory variable robot density on the mediating variable total factor productivity and the joint effect of both robot density and total factor productivity on firms' exports. The results are shown in Table 11.

Columns 1–3 of Table 11 are set in the same way as in Table 10, and we also divide the total effect (TE), direct effect (CDE), and indirect effect (NIE). Table 10 shows that the overall effect of AI and industrial robots on firms' exports is significantly negative, and the direct effect of industrial robots on firms' exports is also significantly negative when the mediating variable total factor productivity is considered, which is also consistent with the results of the all-industry sample benchmark regression in section 4.1. In addition, the significant indirect utility indicates that there is indeed a mediating effect of AI and the use of industrial robots on firms' exports. Combined with the fact that the indirect effect coefficient is significantly positive and the direct effect is significantly negative, it indicates that the effect of the use of industrial

**Table 11.  Regression results of mediating effect.**

|  | (1) | (2) | (3) |
|---|---|---|---|
| **mediating effect** | *op10* | *op10_1* | *lp* |
| direct effect | -0.066*** | -0.046*** | -0.065*** |
|  | (-89.297) | (-101.369) | (-88.548) |
| indirect effect | 0.000*** | 0.004*** | -0.000*** |
|  | (3.526) | (63.674) | (-15.034) |
| total effect | -0.066*** | -0.042*** | -0.065*** |
|  | (-89.231) | (-92.856) | (-88.951) |

robots on the increase of firms' exports through the increase of firms' total factor productivity is a non-complete mediating effect, and although the use of industrial robots promotes the increase of total factor productivity and the increase of total factor productivity also boosts firms' exports, the use of AI and industrial robots does not directly lead to the increase of firms' exports. This finding also indicates a masking effect. That is, the labour substitution effect has a significant offsetting effect on the productivity effect. On the basis of this finding, we conduct further analysis in the following section.

**4.6.2 Factors affecting the predominance of the productivity effect or the substitution effect.** In the previous section, it was found that although the introduction of AI and industrial robots can promote firms' exports through productivity effects, the use of AI and industrial robots eventually show a dampening effect on firms' exports in the all-industry sample. The theoretical hypotheses suggest that this dampening effect is mainly due to the substitution effect caused by industrial robots replacing low-skilled labour, which in turn reflects significant industry heterogeneity. In the following, we test the substitution effect of AI and industrial robots on firms' exports and answer the question, "Which firm characteristics are more conducive to export promotion after the introduction of AI and industrial robots?"

1. Capital–labour ratio

On the basis of the assumption that AI substitutes low-skilled labour and thus leads to a decline in equilibrium wages, Korinek and Stiglitz (2021) constructed a model to illustrate the effect of an economy or country's labour endowment or capital–labour ratio on its economy [3]. Suppose there exist two economies in which a developing country or economy produces labour-intensive intermediate goods and exports to a developed country or economy to trade final consumption goods with an initial labour endowment of $L_d$. The developed country produces final goods, with intermediate goods imported from the developing country and capital $K$ as a factor of production. Assume that both types of economies have their production functions in the form of a Cobb–Douglas production function:

$$Y_i = F(K_i, L_i) = K_i^\alpha L_i^{1-\alpha}, \tag{4}$$

where $i = (A, D)$, $A$ denotes developed economies, and $D$ denotes developing economies. The parameter $\alpha$ denotes the capital factor share, while 1-$\alpha$ is the labour share. On assuming that the price of the final product is a standard unit of quantity, the marginal output of labour in an economy $F_i^L = {}^{(}1 - \alpha)Y_i/_{L_i}$ In turn, it reflects the economy's terms of trade (i.e., the relative prices of exports versus imports).

On assuming that an economy improves labour productivity by introducing AI and industrial robots, as reflected by the growth of parameter $\alpha$ to $d\alpha$ with $d>1$, technological innovation based on replacing low-skilled labour will occur only if the capital–labour ratio in that economy is sufficiently low; that is, it causes labour redundancy:

$$\frac{K_i}{L_i} < \tilde{k} = e^{1/(1-\alpha)}. \tag{5}$$

The effect of technological innovation, mainly in the form of substitution of low-skilled labour, on the marginal output of labour can be expressed as

$$\frac{dF_i^L}{d\alpha} = \frac{d\left[(1-\alpha)^{Y_i}/_{L_i}\right]}{d\alpha} = -\frac{Y_i}{L_i} + \frac{1-\alpha}{L_i} \cdot \frac{dY_i}{d\alpha} = \left[(1-\alpha)ln\left(\frac{K_i}{L_i}\right) - 1\right]\frac{Y_i}{L_i}. \tag{6}$$

If the capital–labour ratio is defined as $k = \frac{K}{L}$, then technological innovation $d\alpha>0$, which takes the main form of substitution of labour, will further worsen the economic situation and

terms of trade of the economy only if the following conditions are met.

$$\frac{K_i}{L_i} < \tilde{k} = k \cdot \frac{1 - (1 - \alpha)lnk}{1 + \alpha \cdot lnk}. \tag{7}$$

According to Eq 7, when an economy (usually a developing economy) has a low capital–labour ratio, the positive productivity effect of technological innovation represented by the use of AI and industrial robots will be lower than the substitution effect of that technological innovation replacing low-skilled labour, thus worsening the economy's terms of trade $F_D^L$, reducing its labour share of income $L_D \cdot F_D^L$, and thus discouraging firms from exporting.

On the basis of the above assumptions, this paper further proposes that the capital–labour ratio of enterprises will have a moderating effect on the substitution effect of industrial robot use affecting firms' exports; for enterprises with a high capital–labour ratio, the substitution of low-skilled labour by industrial robots will not worsen their terms of trade, and the productivity effect will be greater than the negative substitution effect, thus leading to export growth; for enterprises with a low capital–labour ratio and a redundant labour force, the substitution of labour by industrial robots will further worsen their terms of trade, and the negative substitution effect will be greater than the productivity effect, thus inhibiting firms' exports. Improving the capital–labour ratio will help enhance AI's role in promoting firms' exports.

To further verify the moderating effect of the capital–labour ratio, an interaction term between industrial robot density and the firm capital–labour ratio was introduced in this paper. The capital-labour ratio was calculated as the ratio of capital stock to the number of employees. We calculated capital stock using the perpetual inventory method and centralised the robot density and capital–labour ratio variables to construct the interaction term. The regression results are shown in Table 12.

Table 12 reports the regression results for the all-industry sample and the high-technology industry sample. From Table 12, it can be seen that the coefficient of the interaction term between industrial robot density and the capital-labour ratio is significantly positive for the all-industry sample, which verifies that the higher the capital-labour ratio is and the larger the capital stock of the firm is, the less industrial robot use will worsen the terms of trade. It makes the productivity effect larger than the negative substitution effect, and industrial robots can promote firms' exports. The lower the capital–labour ratio, the more industrial robots tend to worsen the terms of trade by replacing labour, thus creating a disincentive for firms to export. However, the moderating effect of the capital–labour ratio is not significant for high-tech industries. This might be because high-skilled workers are mainly engaged in production in high-tech industries. The rise in the capital-labour ratio indicates that more high-skilled labour is replaced by industrial robots. AI cannot significantly promote firms' exports under the condition that high-skilled labour is already scarce. Moreover, the average capital–labour ratio of high-tech industries is already much higher than that of the all-industry sample or non-high-tech industries, so the moderating effect of the capital–labour ratio weakens for high-tech industries (Fig 3). It can also be noticed that the coefficient of robot density is not significant for the all-industry sample after including the interaction term. Considering the coefficient of robot density is significant in the benchmark regression, we believe it shows that the interaction term covers up the main effect of robot density. (At this time, the coefficient of the main effect represents the marginal impact of robot density on firms' exports when the capital–labour ratio equals 0.) Furthermore, the coefficient of robot density is still significantly positive for high-tech industries.

2. Value added of export products

**Table 12. Regression results with the interaction term between industrial robot density and the firm-capital–labour ratio.**

| | All-industry sample | | | High-tech industries | | |
|---|---|---|---|---|---|---|
| | (1) | (2) | (3) | (4) | (5) | (6) |
| **Variable** | *op5_1* | *op10_1* | *op15_1* | *op5_1* | *op10_1* | *op15_1* |
| TFP | 0.464*** | 0.591*** | 0.422*** | 0.529*** | 0.624*** | 0.502*** |
| | (35.625) | (37.697) | (29.097) | (20.788) | (20.389) | (18.183) |
| Tariff | -0.102*** | -0.113*** | -0.113*** | -0.026 | -0.029 | -0.023 |
| | (-5.590) | (-6.029) | (-5.902) | (-0.639) | (-0.685) | (-0.538) |
| Robot density | 0.005 | 0.006 | 0.011 | 0.159*** | 0.151*** | 0.137*** |
| | (0.672) | (0.909) | (1.482) | (8.221) | (7.629) | (6.776) |
| KL | 0.010*** | 0.012*** | 0.010*** | 0.012*** | 0.014*** | 0.019*** |
| | (6.582) | (7.740) | (6.193) | (3.975) | (4.745) | (6.268) |
| Interaction term | 0.002*** | 0.001*** | 0.001*** | 0.001 | 0.001 | -0.000 |
| | (4.646) | (2.865) | (2.747) | (0.790) | (0.620) | (-0.133) |
| Firm Size | 0.079*** | 0.081*** | 0.090*** | 0.090*** | 0.089*** | 0.101*** |
| | (25.982) | (24.963) | (27.111) | (14.991) | (14.079) | (15.376) |
| GDP | 0.018** | 0.025*** | 0.029*** | 0.137*** | 0.139*** | 0.138*** |
| | (2.015) | (2.718) | (3.111) | (8.206) | (8.145) | (7.975) |
| Year FE | Yes | Yes | Yes | Yes | Yes | Yes |
| Importing country FE | Yes | Yes | Yes | Yes | Yes | Yes |
| Product FE | Yes | Yes | Yes | Yes | Yes | Yes |
| Industry FE | Yes | Yes | Yes | Yes | Yes | Yes |
| Firm FE | Yes | Yes | Yes | Yes | Yes | Yes |
| Samples | 6687613 | 6304194 | 6051295 | 2159060 | 2043309 | 1965237 |
| $R^2$ | 0.333 | 0.334 | 0.335 | 0.363 | 0.364 | 0.364 |
| Adjusted $R^2$ | 0.321 | 0.322 | 0.322 | 0.351 | 0.351 | 0.350 |
| $F$-value | 348.042 | 369.277 | 276.677 | 129.796 | 123.267 | 110.041 |

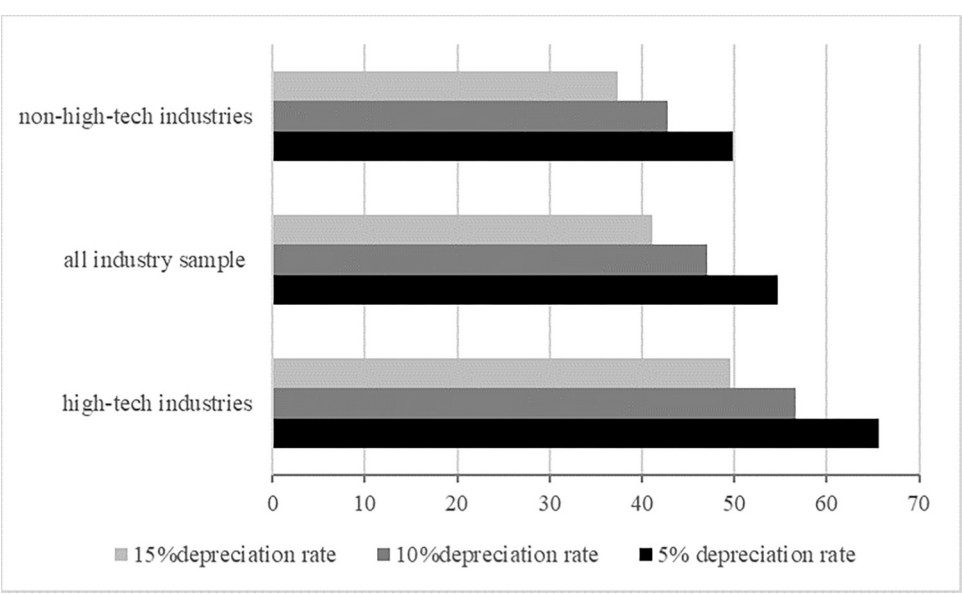

**Fig 3. The average capital–labour ratio of the all-industry sample, high-tech industries, and non-high-tech industries.** Note: The capital–labour ratio was calculated as the ratio of capital stock to the number of employees. We calculated capital stock using the perpetual inventory method with 5%, 10%, and 15% depreciation rates.

To further verify the substitution effect of AI and industrial robots on firms' exports and to demonstrate what factors the substitution effect depends on, on the basis of previous theoretical assumptions, we introduced quality of export products as a proxy variable reflecting product value added, assuming that high-value-added products require more high-skilled labour to produce, while low-value-added products require only low-skilled labour to produce. Therefore, the higher the value added of products produced and exported, the more the requirement for high-skilled labour will be, and the introduction of industrial robots will reflect the productivity effect more and the substitution effect of low-skilled labour less, so the use of industrial robots will eventually lead to an increase in firms' exports. The lower the value added of the products produced and exported by enterprises, the more easily replaceable labour will be. That is, the substitution effect dominates the productivity effect, and the use of AI and industrial robots will lead to a decline in firms' exports. To further illustrate the difference in the impact of industrial robots on firms' exports with different value added, that is, different quality levels, we introduced an interaction term between industrial robots and the quality of the firms' export products and measured the quality of export products using Khandelwal et al.'s method (2013) [33].

Table 13 reports the regression results for the all-industry sample and the high-technology industry, where the three columns correspond to total factor productivity calculated using different depreciation rates. From Table 13, it can be seen that the coefficient of the interaction term between robot density and export product quality is significantly positive for both the all-industry sample and high-tech industries. That is, the higher the export product quality and the higher the product value added, the more AI and industrial robots will tend to promote exports, and the more the productivity effect will dominate the negative substitution effect, which again verifies the theoretical hypotheses. This finding proves that the added value of the product, that is, the quality of the export product, is an important factor in determining whether AI and industrial robots will promote firms' exports. The coefficient of robot density is still significantly negative for the all-industry sample, while it is significantly positive for high-tech industries, and this finding is consistent with the previous empirical analysis results.

## 5. Conclusions and policy implications

This paper used industry-level IRF industrial robot data with Chinese micro-enterprise data to empirically test the mechanisms by which AI and industrial robots affect firms' exports. The following conclusions can be drawn from the findings:

1. This paper parsed the impact of AI and industrial robots on firms' exports as a result of the joint effect of productivity effect and labour substitution effect and found through empirical tests that although the use of industrial robots promotes productivity, the use of industrial robots had an inhibiting effect on the exports of Chinese industrial enterprises from 2002 to 2013. That is, the negative substitution effect was dominant over the positive productivity effect, showing certain developing country characteristics.

2. The impact of AI and industrial robots on firms' exports in China shows obvious industry heterogeneity. For high-tech industries, the use of industrial robots does promote firms' exports because the proportion of low-skilled labour is relatively low and because the productivity effect dominates the substitution effect. However, for most non-high-tech industries, the use of industrial robots does not play a role in promoting firms' exports.

3. The impact of AI and industrial robots on firms' exports in China also shows some temporal variability, and we found two turning points through empirical analysis: One was in 2003, before which the use of industrial robots mainly showed an inhibiting effect on firms' exports, which was gradually reflected as a driving effect thereafter; the second was in 2006,

**Table 13. Regression results with the interaction term between industrial robot density and export product quality.**

| | All-industry sample | | | High-tech industries | | |
|---|---|---|---|---|---|---|
| | (1) | (2) | (3) | (4) | (5) | (6) |
| Variable | op5_1 | op10_1 | op15_1 | op5_1 | op10_1 | op15_1 |
| TFP | 0.211*** | 0.267*** | 0.156*** | 0.254*** | 0.297*** | 0.173*** |
| | (31.504) | (32.026) | (20.620) | (17.341) | (16.381) | (10.762) |
| Tariff | -0.149*** | -0.154*** | -0.155*** | -0.361*** | -0.360*** | -0.351*** |
| | (-13.571) | (-13.595) | (-13.430) | (-13.346) | (-12.952) | (-12.399) |
| Robot Density | -0.070*** | -0.070*** | -0.067*** | 0.236*** | 0.233*** | 0.225*** |
| | (-17.649) | (-17.095) | (-15.926) | (19.596) | (18.794) | (17.791) |
| Product Quality | 12.585*** | 12.576*** | 12.561*** | 11.950*** | 11.953*** | 11.951*** |
| | (2691.757) | (2603.928) | (2541.376) | (1312.376) | (1279.898) | (1256.088) |
| Interaction term | 0.014*** | 0.019*** | 0.021*** | 0.378*** | 0.386*** | 0.381*** |
| | (8.144) | (10.793) | (11.515) | (86.120) | (86.020) | (83.254) |
| Firm Size | 0.020*** | 0.022*** | 0.026*** | 0.021*** | 0.019*** | 0.025*** |
| | (10.998) | (11.445) | (13.381) | (5.461) | (4.625) | (5.942) |
| GDP | -0.106*** | -0.104*** | -0.101*** | -0.088*** | -0.087*** | -0.083*** |
| | (-18.791) | (-18.106) | (-17.284) | (-7.897) | (-7.673) | (-7.242) |
| Year FE | Yes | Yes | Yes | Yes | Yes | Yes |
| Importing country FE | Yes | Yes | Yes | Yes | Yes | Yes |
| Product FE | Yes | Yes | Yes | Yes | Yes | Yes |
| Industry FE | Yes | Yes | Yes | Yes | Yes | Yes |
| Firm FE | Yes | Yes | Yes | Yes | Yes | Yes |
| Samples | 6641530 | 6261133 | 6010608 | 2129137 | 2015104 | 1938623 |
| $R^2$ | 0.747 | 0.747 | 0.747 | 0.718 | 0.718 | 0.718 |
| Adjusted $R^2$ | 0.743 | 0.742 | 0.742 | 0.712 | 0.712 | 0.712 |
| $F$-value | 1.1e+06 | 1.0e+06 | 9.7e+05 | 2.6e+05 | 2.5e+05 | 2.4e+05 |

before which the impact of industrial robots on firms' exports was not significant and after which the use of industrial robots began to significantly promote firms' exports.

4. This study further explored the question of what kind of enterprises are more likely to benefit from the use of AI and industrial robots and found that the higher the capital-to-labour ratio, the easier it is for enterprises to benefit from the introduction of industrial robots.

Accordingly, this paper proposes the following policy recommendations: First, industrial transformation efforts should be increased to turn traditional manufacturing industries into high-tech industries depending on the productivity dividend brought by artificial intelligence and industrial robots to promote firms' exports. Second, enterprises should be encouraged to produce high-quality, high-value-added products, which will contribute to boosting productivity. Relying on the double improvement of increased productivity and product quality, China will further expand into overseas markets and gain a greater competitive advantage in the international market. Third, the proportion of the low-skilled labour force should be reduced to improve the quality of enterprise labour forces. Moreover, the capital–labour ratio should be improved in enterprises while realising the dual transformation and upgrade of industrial structure and comparative advantage in international trade to comprehensively improve the quality and skill level of the manufacturing labour force through education, training, and so forth. The basis of China's manufacturing industry making full use of the AI dividend is to improve the quality of the labour force and reduce the proportion of the low-skilled labour force.

## Author Contributions

**Conceptualization:** Zhaozhong Zhang.

**Data curation:** Zhaozhong Zhang.

**Formal analysis:** Zhaozhong Zhang.

**Funding acquisition:** Zhaozhong Zhang.

**Investigation:** Zhaozhong Zhang.

**Methodology:** Zhaozhong Zhang.

**Software:** Zhaozhong Zhang.

**Supervision:** Zhaozhong Zhang.

**Validation:** Zhaozhong Zhang.

**Writing – original draft:** Zhaozhong Zhang.

**Writing – review & editing:** Zhaozhong Zhang, Fangfang Deng.

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
