## [Decision Letter · Decision Letter 0]

13 Dec 2022

PONE-D-22-32882How artificial intelligence becomes the growth power of firms' export? Evidence from ChinaPLOS ONE

Dear Dr. Zhang,

Thank you for submitting your manuscript to PLOS ONE. After careful consideration, we feel that it has merit but does not fully meet PLOS ONE’s publication criteria as it currently stands. Therefore, we invite you to submit a revised version of the manuscript that addresses the points raised during the review process.

I recommend that it should be revised taking into account the changes requested by the reviewers. Since the requested changes include valuable and constructive reviews, I would like to give you a chance to revise your manuscript. The revised manuscript will undergo the next round of review by same reviewers.

We look forward to receiving your revised manuscript.

Kind regards,

Baogui Xin, Ph.D.

Academic Editor

PLOS ONE

Journal Requirements:

2. During your revisions, please confirm whether the wording in the title is correct and update it in the manuscript file and online submission information if needed. Specifically, consider removing the question mark, and/or rewording the whole title so that is both grammatically correct and easy to understand.

Reviewers' comments:

Reviewer's Responses to Questions

**Comments to the Author**

1. Is the manuscript technically sound, and do the data support the conclusions?

Reviewer #1: Yes

Reviewer #2: Yes

Reviewer #3: Yes

2. Has the statistical analysis been performed appropriately and rigorously? 

Reviewer #1: I Don't Know

Reviewer #2: No

Reviewer #3: Yes

3. Have the authors made all data underlying the findings in their manuscript fully available?

Reviewer #1: No

Reviewer #2: Yes

Reviewer #3: Yes

4. Is the manuscript presented in an intelligible fashion and written in standard English?

Reviewer #1: Yes

Reviewer #2: No

Reviewer #3: No

5. Review Comments to the Author

Reviewer #1: Please provide all the data that supports these findings.

Very nice written introduction with clearly presented references.

Moreover, There are a few suggestions

1.Literature reviews can be more structured

2.Consider the numerical optimization of Table 1,to be more visible

3.The article does not yet reflect the point of innovation

4.Note the format of the references

“The authors based on Evidence from China divides the impact mechanism into productivity effect and labor

substitution effect” The manuscript title,abstract and conclusion is very clear and presented well.The introduction gives a thorough description,followed by methodology adopted and discussions of the results.

In my opinion,the manuscript is well formatted and written well with a significant work and hence I would suggest acceptance in the present form.

Reviewer #2: By using Chinese Custom data, the Chinese Industrial Firm data and the robot data from the International Robot Federation (IRF)，the manuscript discussed the impact of industrial robots on firms' export value. Then, the manuscript conducted mechanism and heterogeneity analysis. The perspective of this research is interesting and innovative. However, the author should further elaborate the paper. As such I suggest the following suggestions.

1.Abstract. The authors argue that “The impact of artificial intelligence on firms' export value also varies by time period”. However, what are the different impacts? It needs to be clarified.

2. Introduction. There are relatively few related discussions on the impact of AI and industrial robots on firms' export, which needs to be further strengthened to highlight the necessity of this study.

3.The contributions. The contributions in the manuscript are like explaining the research content, rather than contributions. The author's contributions need to be further refined.

4. On page 9, the chapter arrangement does not correspond to the article, check it.

5. Theoretical hypotheses.

1) The hypothesis to be tested in the paper (such as H1, H2, H3, etc.) should be clearly pointed out. 2) The paper demonstrates the effect of time heterogeneity in the empirical analysis, but this is not shown in the theoretical hypothesis part. It is suggested to supplement.

6.Data.

On page 11, “In this paper, only the data of this database from 2002 to 2013 are selected, mainly because it is difficult to estimate the total factor productivity of enterprises due to the lack of statistics of industrial value added, intermediate inputs and other indicators in the data of Chinese industrial enterprises after 2007, and there are problems of inconsistency in the selection criteria of sample enterprises and the statistical caliber of variables”.

The article mentioned the lack of industrial value added and there are problems of inconsistency in the selection criteria of sample enterprises and the statistical caliber of variables after 2007, but the author still chose the data from 2007 to 2013. I cannot understand the intention of the article. Besides, the data of 2002-2007 and 2002-2013 are all used in the following empirical analysis. So, I suggest you explain the sample time span here clearly.

7. In the article, both OP and LP methods are used to calculate TFP, but which measurement method was used in Table 2 has not been explained.

8. When measuring enterprise’s TFP with LP, it is suggested that capital input, labor input and intermediate input variables should be supplemented.

9. All empirical regression tables are not standardized and lack of annotations. The meaning of the value in brackets and the significance level of the asterisk is not explained, although it does not affect the understanding of the empirical results.

10. On Page 18, the interpretation of the results in Table 3 does not correspond to the contents of the table, the author needs to check it carefully.

11.Table 3.

1) The results in columns (3) and (4) in Table 3 are not significant, and the coefficient in column (5) is significantly positive, which is inconsistent with the benchmark analysis. Can it indicate that the results are robust?

2) It is suggested to add robustness testing methods to enhance the reliability of the results. For example, change the estimation sample, change the variable measurement method, etc.

12. On page 20, “2013 data are positively correlated at a significance level of 10%, which again reflects that the impact of AI on the export behavior of micro-enterprises in China has some temporal differences”. I don't think this explanation is reasonable. The results are based on the sample from 2002 to 2013, not 2013 data. The author should give reasonable reasons.

13.IV regression. In Table 8 and Table 9, the coefficients of columns (1) and (3) are negative, but not significant, which means that the benchmark regression results are not robust after considering the endogenous problem, and the reliability of the conclusion is worrying.

14. Mechanism test. This paper verifies the factors affecting the productivity effect and the substitution effect through interaction term, but single variables also need to be added. That is, in Table 12 industrial robot density and capital-labor ratio should add in the regression model. And, in Table 13 robot density and export products quality should add in the model. The author needs to re-estimate the results in Table 12 and Table 13.

15. At last, there are a lot of expression and typo errors, please revise and check the whole manuscript carefully. It is suggested to polish the language of the article.

Reviewer #3: Reconsideration after major revision.

The study explores the impact of artificial intelligence and industrial robots on firms' export behavior and divides the impact mechanism into productivity effect and labor substitution effect.

This paper examines the effect of industrial robots on firms' export value by using Chinese Custom data, the Chinese Industrial Firm data and the robot data from the International Robot Federation (IRF).

However, the paper is not yet suitable for publication in "PLOS ONE" and requires major revisions.

1.  The contributions should be strengthened.

2.  The models used in the paper are not visualized, a figure is recommended.

A remark and discussion on the orientation of the model is necessary.

3.  Variable selection:

The reasons of variables selection are strongly proposed to present.

4.  It is unclear how the results presented can be used to support the decision makers.

There are many SERIOUS spelling and style problems in the paper, such as:

(1)The initial letters of the key words should be capitalized.

(2) For Chinese Customs, the ‘s’ is necessary.

(3) Reference style.

(4) Numbering.

(5) Brynjolfsson and McAfee (2014) referred to AI and the digital revolution as the second machine revolution, arguing that it is as important as the industrial revolution in driving development [4].

Brynjolfsson E, Mcafee A . The Second Machine Age[J]. Nz Business, 2014, 14(11):1895-1896.

6. PLOS authors have the option to publish the peer review history of their article (what does this mean?). If published, this will include your full peer review and any attached files.

Reviewer #1: No

Reviewer #2: No

Reviewer #3: No

---

## [Author Response · Author response to Decision Letter 0]

24 Jan 2023

Thanks very much for the reviewers' comments concerning our manuscript entitled “How can artificial intelligence boost firms' exports? Evidence from China” (Manuscript Number PONE-D-22-32882). The comments are valuable and helpful for improving our paper. We have studied the comments carefully and made corrections according to the suggestions. A revised manuscript with the modified sections marked in red is attached as supplemental material for ease of checking. The details of responses to reviewers are shown in the file 'Response to Reviewers'.

---

## [Decision Letter · Decision Letter 1]

17 Feb 2023

PONE-D-22-32882R1How can artificial intelligence boost firms' exports? Evidence from China

PLOS ONE

Dear Dr. Zhang,

Thank you for submitting your manuscript to PLOS ONE. After careful consideration, we feel that it has merit but does not fully meet PLOS ONE’s publication criteria as it currently stands. Therefore, we invite you to submit a revised version of the manuscript that addresses the points raised during the review process.

Please polish your manuscript as asked by Reviewer #3. The Academic Editor will only review the manuscript in the next round to speed the review process.

We look forward to receiving your revised manuscript.

Kind regards,

Baogui Xin, Ph.D.

Academic Editor

PLOS ONE

Journal Requirements:

Reviewers' comments:

Reviewer's Responses to Questions

**Comments to the Author**

1. If the authors have adequately addressed your comments raised in a previous round of review and you feel that this manuscript is now acceptable for publication, you may indicate that here to bypass the “Comments to the Author” section, enter your conflict of interest statement in the “Confidential to Editor” section, and submit your "Accept" recommendation.

Reviewer #2: All comments have been addressed

Reviewer #3: All comments have been addressed

2. Is the manuscript technically sound, and do the data support the conclusions?

Reviewer #2: Yes

Reviewer #3: Yes

3. Has the statistical analysis been performed appropriately and rigorously? 

Reviewer #2: Yes

Reviewer #3: Yes

4. Have the authors made all data underlying the findings in their manuscript fully available?

Reviewer #2: Yes

Reviewer #3: Yes

5. Is the manuscript presented in an intelligible fashion and written in standard English?

Reviewer #2: Yes

Reviewer #3: No

6. Review Comments to the Author

Reviewer #2: The author made detailed modifications to my comments. I am very satisfied with these modifications.

Reviewer #3: The language of the paper still has problems of expression and grammatical errors. The author is requested to revise the language carefully to make the language correct, clear and in line with academic norms.

7. PLOS authors have the option to publish the peer review history of their article (what does this mean?). If published, this will include your full peer review and any attached files.

Reviewer #2: No

Reviewer #3: No

---

## [Author Response · Author response to Decision Letter 1]

6 Mar 2023

Responses to Reviewers

Thanks very much for the reviewers’ comments concerning our manuscript entitled “How can artificial intelligence boost firms' exports? Evidence from China” (Manuscript Number PONE-D-22-32882). The comments are very helpful for improving our paper. A revised manuscript with the modified sections marked in red is attached as supplemental material for ease of checking. 

The responses and modifications are as follows:

Reviewer 2

The author made detailed modifications to my comments. I am very satisfied with these modifications.

Response to comment:

Thanks for the comment. The reviewer’s suggestions are very helpful for us to improve the paper. 

Reviewer 3

The language of the paper still has problems of expression and grammatical errors. The author is requested to revise the language carefully to make the language correct, clear and in line with academic norms.

Response to comment:

Thanks for the comment and suggestion. We have carefully checked the problems of expression and grammatical errors in this paper. We have also polished the language of the article again and revised the manuscript carefully. The reviewer can see this in the revised manuscript.

---

## [Editor Report · Decision Letter 2]

6 Mar 2023

How can artificial intelligence boost firms' exports? Evidence from China

PONE-D-22-32882R2

Dear Dr. Deng,

We’re pleased to inform you that your manuscript has been judged scientifically suitable for publication and will be formally accepted for publication once it meets all outstanding technical requirements.

Kind regards,

Baogui Xin, Ph.D.

Academic Editor

PLOS ONE
---

## [Editor Report · Acceptance letter]

9 May 2023

PONE-D-22-32882R2 

How can artificial intelligence boost firms’ exports? Evidence from China 

Dear Dr. Deng:

I'm pleased to inform you that your manuscript has been deemed suitable for publication in PLOS ONE. Congratulations! Your manuscript is now with our production department. 

Kind regards, 

on behalf of

Professor Baogui Xin 

Academic Editor

PLOS ONE